



# The impact of measurement precision on the resolvable resolution of ice core water isotope reconstructions

Fyntan Shaw[1], Thomas Münch[1], Vasileios Gkinis[2], and Thomas Laepple[1,3]

[1]Alfred Wegener Institute, Helmholtz Centre for Polar and Marine Research, Potsdam, Germany
[2]Niels Bohr Institute, Physics of Ice, Climate, and Earth, University of Copenhagen, Copenhagen, Denmark
[3]University of Bremen, MARUM - Centre for Marine Environmental Sciences and Faculty of Geosciences, Bremen, Germany

**Correspondence:** Fyntan Shaw (fyntan.shaw@awi.de)

**Abstract.**

Stable water isotopes in ice cores serve as a valuable proxy for the climate of the past hundreds of thousands of years. Over time, water isotope diffusion causes significant attenuation of the isotopic signal, exacerbated in deep ice due to extreme layer thinning and increased temperatures from geothermal heat flux. This damping affects higher frequencies to a greater extent, erasing information on the shortest timescales. It is possible to restore some of the attenuated variability through deconvolution, a method which reverses the effect of diffusion. However, since the measured isotopic signal always contains noise from the measurement process, deconvolution inevitably amplifies this measurement noise along with the isotopic signal. Thus the effectiveness of deconvolution depends on the precision of the measurements, with noisier data limiting the ability to restore otherwise resolvable frequencies. Here, we quantify the upper frequency limit introduced by the magnitude of the measurement noise analytically for different climate states, and offer a numerical example using the Beyond EPICA Oldest Ice Core (BE-OIC). We also demonstrate the qualitative significance of measurement noise on simulated Antarctic isotopic profiles. The general resolution improvement for firn or upper ice records is on the order of 1.5 times for a 10-fold reduction in measurement noise. Similarly, throughout the BE-OIC, we find the deconvolution of $\delta^{18}$O records with measurement error of 0.1‰ contributes a 1.5 times increase in the maximum resolvable frequency, which rises to a factor of 2 improvement after reducing the measurement noise to 0.01‰. While progress is continuously being made towards improving precision of stable isotope measurements, further improvements using longer integration times should be considered when analysing limited and precious deep ice in order to obtain the most faithful climate reconstructions possible.

## 1 Introduction

Ice cores offer a valuable archive of climate proxies, including stable water isotopes ($\delta^{18}$O, $\delta$D, $\delta^{17}$O) which are known to relate to air temperature (Dansgaard, 1964). Thus, the drilling of deep ice cores enables the retrieval of long, continuous temperature proxy profiles hundreds of thousand of years in the past (EPICA community members, 2004; NEEM community members, 2013; NGRIP members, 2004; Petit et al., 1999). However, over time these isotopic records are subject to alteration, primarily due to the constant motion of the archived water molecules, a process known as diffusion, and quantified by the diffusion length (Johnsen, 1977; Johnsen et al., 2000). This has the undesirable effect of reducing high frequency variability, smoothing the



signal. Diffusion occurs both in the porous firn through vapour-snow exchange (Johnsen, 1977; Whillans and Grootes, 1985), and in the solid ice matrix through ice diffusivity (Ramseier, 1967; Nye, 1998). While the former process ceases below the pore close-off depth, the slower ice diffusion is amplified in the warmer, deeper parts. This becomes increasingly problematic for deep ice cores, as due to the age, temperature and extreme thinning of the ice, the smoothing effect of the ice diffusion is strongest at the bottom of the ice sheet, where the oldest climate information is stored. Methods to retrieve this original signal

are thus crucial for the use of water isotopes in deep ice cores for paleoclimate reconstructions.

     A commonly employed technique for signal recovery is deconvolution, a method capable of reversing the effects of a linear impulse response on a desired signal. When applied to water isotope diffusion, deconvolution is often referred to as back-diffusion, and the impulse response takes the form of a Gaussian (Johnsen et al., 2000). If the variability of the initial, desired

signal and the magnitude of diffusion are known, the attenuated frequencies can be amplified, restoring their original variability. However, measured records also have noise introduced by the measurement process. For strongly diffused ice, the measurement noise will mask the climate information at the highest frequencies, so the full restoration of these frequencies through deconvolution is not possible without greatly inflating the measurement noise. Therefore, a compromise must be found between recovering the climate signal and avoiding amplification of the measurement noise. Wiener deconvolution offers an

optimal solution, by calculating the amplification factor for each frequency which minimises the mean square error between the initial and deconvolved records (Wiener, 1949; Johnsen, 1977). This method cannot improve the signal-to-noise ratio (hereafter SNR) of any frequencies, and so it is optimal to dampen the variations at frequencies where measurement noise dominates the signal. Thus, there is an effective limit on the highest resolvable frequency through Wiener deconvolution that depends on the SNR and the diffusion length. Reducing measurement noise would, in turn, enable the recovery of higher frequencies that were

previously irretrievable (Fig. 1).





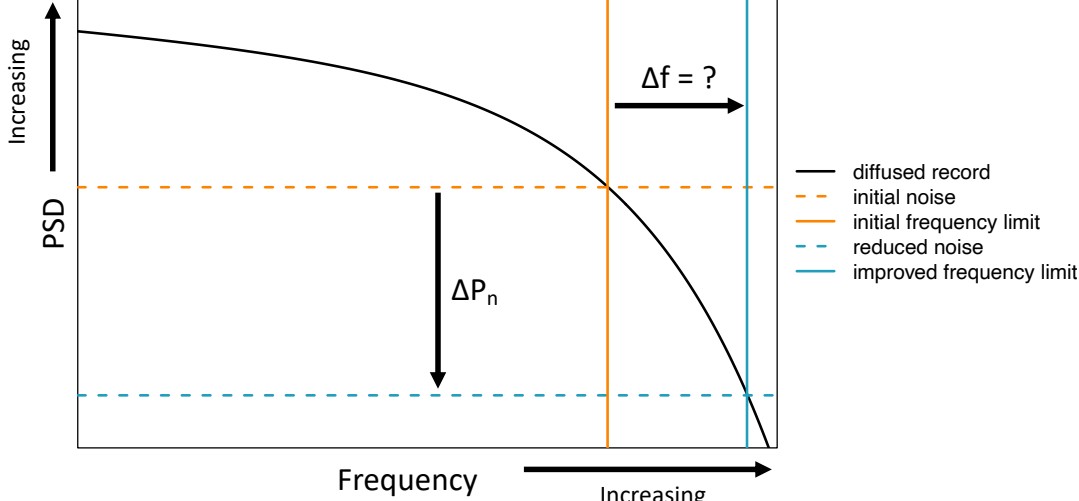

**Figure 1.** Conceptual diagram illustrating how reducing the measurement noise will enable the recovery of higher frequencies. Shown is an idealised power spectrum of a diffused signal (black), and measurement noise (dashed horizontal lines). Reducing the measurement noise (orange to blue) increases the frequency at which SNR = 1 (solid vertical lines).

We are particularly interested in the effect on the Beyond EPICA - Oldest Ice Core (hereafter BE-OIC) project, due to its aim in acquiring a climate record dating back 1.5 million years, unprecedented in ice core records. In these deepest sections, millennial variability will likely be attenuated beyond recovery, and even the 41,000 year (41 kyr) glacial-interglacial cycles

before the Mid-Pleistocene Transition could be dampened. It is therefore crucial to be aware of the limitations introduced by our measurement process so attempts can be made to recover the sought-after temporal resolution.

In this paper, we determine the relationship between measurement precision and highest resolvable frequency. We begin by addressing the problem analytically, deriving general equations that describe the highest resolvable frequency and its depen-

dencies across various scenarios. We then apply these equations to the Beyond EPICA – Oldest Ice Core project, estimating the measurement precision required to achieve specific temporal resolutions, particularly in the deepest and oldest sections of the core. We simulate surrogate BE-OIC records with differing levels of measurement noise, and apply Wiener deconvolution to visualise the qualitative improvement offered by very precise data. Finally, we discuss potential techniques for achieving such high precision through long-integration time measurements.

## 2 Data and Methods

### 2.1 Data

For our application to the BE-OIC, we use the age-depth model from Chung et al. (2023), based on a 1D numerical model, radargrams of the Dome C area and the EPICA Dome C (EDC) ice core age profile. This model is used for the transformation of





frequencies between the time domain and the depth domain in the following analysis. We also use the discrete 11 cm resolution
$\delta^{18}O$ data from the EDC ice core (EPICA community members, 2004; Grisart et al., 2022) to estimate the isotopic variability
before diffusion in the Dome C area.

## 2.2 Power spectrum describing the isotopic variability at Dome C

Over deep ice cores such as the BE-OIC we are working with vastly different frequencies at different depths, spanning decadal
to multi-millennial timescales. Before diffusion, water isotope records from ice cores are commonly approximated as white-
noise on sub-annual to centennial timescales (Johnsen et al., 2000; Gkinis et al., 2014; Jones et al., 2017) justified by precipita-
tion intermittency and post-depositional processes obfuscating the temperature signal (Fisher et al., 1985; Johnsen et al., 1997;
Münch and Laepple, 2018; Casado et al., 2020). However, these processes have a lesser effect on longer timescales, which tend
to display greater low-frequency variability due to a relatively stronger climate signal. Instead, the power spectra of records on
longer timescales often closely resemble a power-law (Pelletier, 1998; Huybers and Curry, 2006; Laepple and Huybers, 2014;
Shaw et al., 2024). A simple model across all frequencies for the power spectrum before diffusion (hereafter $P_0$) of isotopic
records from the Dome C region is therefore a combination of the models of both regimes,

$$P_0(f) = \alpha f^{-\beta} + \epsilon, \tag{1}$$

where $\alpha f^{-\beta}$ is the low frequency power-law component characterised by numerical constants $\alpha$ and $\beta$, and $\epsilon$ is a constant
representing the high frequency white-noise component.


To best estimate $P_0$, we use $\delta^{18}O$ data from the current oldest ice core to have been retrieved, the EDC ice core. We bin the
data to equidistant spacing in time which is required to compute the power spectrum. Since we are interested in all timescales,
we create multiple timeseries using different bin sizes. We then estimate the power spectrum of each of these timeseries
using Thomson's multitaper method (Percival and Walden, 1993) and compute the mean across all individual spectra (see
Appendix A). Using Eqn. 1 we find a best fit in logarithmic space, to attribute proportional weighting to all frequencies (Fig.
2). Frequencies greater than 10 kyr$^{-1}$ were excluded due to the influence of the firn diffusion, while frequencies less than 0.01
kyr$^{-1}$ were also excluded due to the known bias in the spectral estimation method.





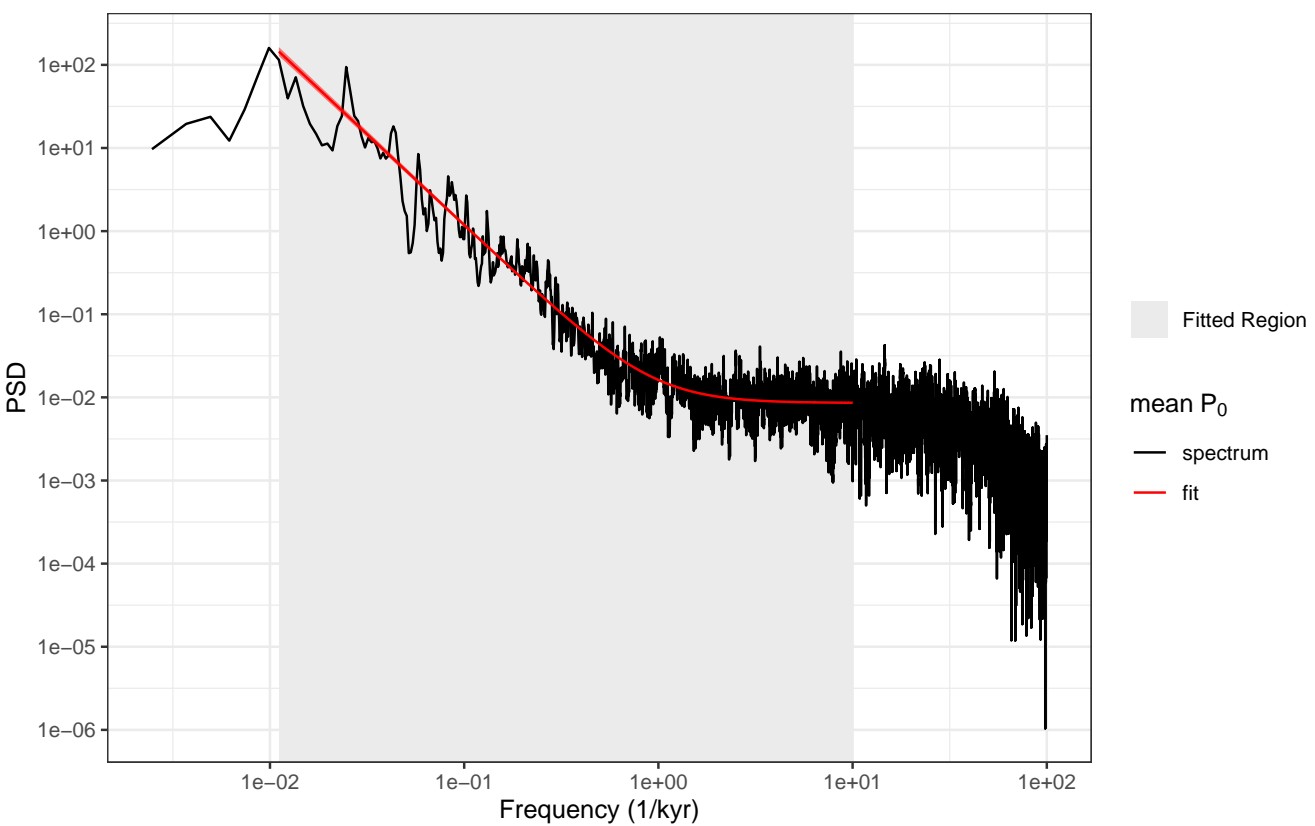

**Figure 2.** $P_0$ fit (red) over the grey shaded region of a composite power spectrum of the EDC water isotope record (black). By excluding high frequencies attenuated by firn diffusion, this fit serves as a good approximation of the initial isotopic variability of the BE-OIC

## 3 Theory

### 3.1 Diffusion and Wiener deconvolution

The effect of diffusion on some isotopic profile $x(z)$ is mathematically represented as a convolution of the profile with a Gaussian filter $h(z)$,

$$h(z) = \frac{1}{\sigma\sqrt{2\pi}} e^{\frac{-z^2}{2\sigma^2}}. \tag{2}$$

Here, $\sigma$ represents the diffusion length, in units equal to the depth $z$. The final measured record $y(z)$ including some measurement noise term $n(z)$ is given by,

$$y(z) = x(z) * h(z) + n(z), \tag{3}$$





where $*$ is the convolution operator. Due to the convolution theorem (Cochran et al., 1967), it is easier to work in the Fourier domain,

$$Y(f) = X(f)H(f) + N, \tag{4}$$

where capitalised variables represent the corresponding Fourier transforms. Taking the absolute square of the Fourier trans-
forms gives us the power spectra,

$$P(f) = P_0(f)e^{(-2\pi f \sigma)^2} + P_n, \tag{5}$$

where $P(f)$ and $P_0(f)$ are the power spectrum of $y(z)$ and $x(z)$ respectively, $f$ represents frequency in the reciprocal units
of $z$ and $P_n(f)$ is the power spectrum of the measurement noise. To recover the best estimate, $\hat{x}(z)$, of the original isotopic
signal $x(z)$, we want to find some function $g(z)$ that can reverse the effects of diffusion without inflating the measurement
noise. After convolving our measured record $y(z)$ with this function,

$$\hat{x}(z) = g(z) * y(z), \tag{6}$$

we want the mean square error between $x(z)$ and our estimate $\hat{x}(z)$ to be minimised. The function $g(z)$ that provides such
an estimate is the Wiener filter (Wiener, 1949), which has the Fourier transform,

$$G(f) = \frac{P_0(f)H^*(f)}{P_0(f)|H(f)|^2 + P_n(f)}. \tag{7}$$

Using the convolution theorem and the inverse Fourier transform $\mathcal{F}^{-1}$, $\hat{x}(z)$ can be calculated from

$$\hat{x}(z) = \mathcal{F}^{-1}(G(f)Y(f)). \tag{8}$$

From Eqn. 7, if $P_n(f) = 0$ then the Wiener filter simplifies to $G(f) = \frac{1}{H(f)}$, and completely reverses the smoothing effect,
restoring the exact original record. In reality, the noise term reduces the amplification of the diffused frequencies, with a greater
reduction the more the noise dominates the diffused signal. Wiener deconvolution balances the recovery of the desired signal
and the suppression of over-amplifying undesired noise, such that the mean square error is minimised.

### 3.2 Highest resolvable frequency

To determine what effect measurement precision has on the recoverable resolution, we first need to define the highest resolvable
frequency $f_{\max}$. Intuitively, we can define a frequency as resolvable if its amplitude is above some fraction of its initial,



undiffused amplitude. We call this the relative amplitude, $A$. For a diffused isotopic record, the relative amplitude $A_{\mathrm{diff}}$ is simply the diffusion transfer function,

$$A_{\mathrm{diff}} = H(f) = e^{(-2\pi f \sigma)^2} \tag{9}$$

while for a deconvolved isotopic record, the relative amplitude $A_{\mathrm{decon}}$ will also depend on the Wiener filter (see Fig. 3),

$$A_{\mathrm{decon}} = H(f)G(f) = \frac{P_0(f)|H(f)|^2}{P_0(f)|H(f)|^2 + P_n(f)} \tag{10}$$

The corresponding highest resolvable frequencies ($f_{\mathrm{max\_diff}}$, $f_{\mathrm{max\_decon}}$) are thus the frequencies at which the relative amplitude drops to some minimum value (Fig. 3),

$$A_{\mathrm{diff\_min}} = H(f_{\mathrm{max\_diff}}) \tag{11}$$

$$A_{\mathrm{decon\_min}} = H(f_{\mathrm{max\_decon}})G(f_{\mathrm{max\_decon}}) \tag{12}$$

Eqn. 11 can be rearranged for $f_{\mathrm{max\_diff}}$,

$$f_{\mathrm{max\_diff}} = \frac{1}{\pi\sigma}\sqrt{-\frac{\ln A_{\mathrm{diff\_min}}}{2}} \tag{13}$$

with dependency only on diffusion length and our chosen minimum relative amplitude.

For the deconvolved record, there are additional dependencies on the initial signal $P_0(f)$ and the measurement noise $P_n(f)$. In this study, we take $P_n$ to be constant with no frequency dependence, a reasonable assumption for discrete measurements as the noise between measurements should be uncorrelated. For the initial signal, we look at two cases. Firstly, we consider firn records or ice cores retrieved from areas with high stratigraphic noise, for which we can also approximate $P_0$ as white-noise with no frequency dependence (Johnsen et al., 2000; Shaw et al., 2024). Given this assumption, we can rearrange Eqn. 12 for the maximum resolvable frequency,

$$f_{\mathrm{max\_decon}} = \frac{1}{2\pi\sigma}\sqrt{\ln\frac{P_0\left(\frac{1}{A_{\mathrm{decon\_min}}} - 1\right)}{P_n}} \tag{14}$$

Reducing the power spectrum of the noise by a factor $\eta$ results in the same $A_{\mathrm{decon\_min}}$ at some new frequency $f'_{\mathrm{max\_decon}}$





$$f'_{\text{max\_decon}} = \frac{1}{2\pi\sigma}\sqrt{\ln\frac{\eta P_0\left(\frac{1}{A_{\text{decon\_min}}}-1\right)}{P_n}} \tag{15}$$

The relative gain in frequency, $f_{\text{gain}}$, from this noise reduction $\eta$ is given by the ratio of these two frequencies,

$$f_{\text{gain}} = \frac{f'_{\text{max\_decon}}}{f_{\text{max\_decon}}} = \sqrt{1+\frac{\ln\eta}{\ln\frac{P_0\left(\frac{1}{A_{\text{decon\_min}}}-1\right)}{P_n}}} \tag{16}$$

For the long time scales of deep ice cores, the white-noise $P_0$ assumption is no longer applicable (Shaw et al., 2024). In this

case, the climate signal will have some frequency dependence and thus rearranging for frequency is not analytically solvable. However, we can still rearrange Eqn. 12 to give the measurement noise required for a specific frequency,

$$P_n(f_{\text{max\_decon}}, A_{\text{decon\_min}}) = P_0(f_{\text{max\_decon}})e^{-(2\pi f_{\text{max\_decon}}\sigma)^2}\left(\frac{1}{A_{\text{decon\_min}}}-1\right) \tag{17}$$

Notably, if we aim to restore 50% of the amplitude of a given frequency, i.e. $A_{\text{decon\_min}} = 0.5$, the noise required would be

$$P_n = P_0(f_{\text{max\_decon}})e^{-(2\pi f_{\text{max\_decon}}\sigma)^2} \tag{18}$$

which is also the noise required for a SNR of 1 (first RHS term = second RHS term from Eqn. 5). In the following, we take $A_{\text{diff\_min}} = A_{\text{decon\_min}} = 0.25$ as the lowest amplitude which we consider recovered. This means at our highest resolvable frequency there is a 75% loss of the initial signal amplitude.





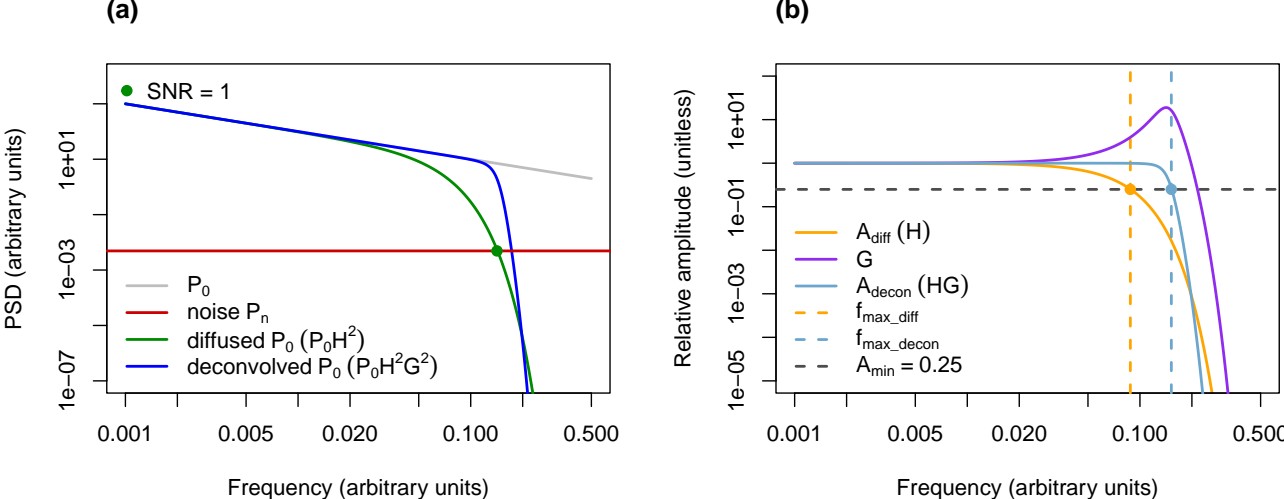

**Figure 3.** Concept of diffusion and Wiener deconvolution in the frequency domain, and the definition of $f_{\max}$. **(a)** The power spectra of the isotopic signal before (grey) and after (green) diffusion, and after deconvolution (blue). The frequency at which the diffused signal intersects the measurement noise (i.e. SNR = 1) is shown by the green dot. **(b)** The Gaussian diffusion filter (orange) and Wiener filter (purple) in the Fourier domain. The highest resolvable frequency of the diffused record $f_{\max\_diff}$ (dashed orange) and deconvolved record $f_{\max\_decon}$ (dashed blue) occur where the corresponding relative amplitudes $A_{\text{diff}}$ and $A_{\text{decon}}$ intersect the $A_{\min} = 0.25$ line (dashed black).

# 4 Results

## 4.1 Generalised white-noise case

We first investigate how a reduction in measurement noise influences the maximum resolvable frequency for the case where $P_0$ is approximated as white-noise. For this, we adopt a value of $A_{\min\_decon} = 0.25$ and three different $\frac{P_0}{P_n}$ ratios (hereafter SNR$_0$) of 5, 10 and 20, and analyse the gain in resolvable frequency, $f_{\text{gain}}$, following Eqn. 16 (Fig. 4). We see a non-linear relationship between $f_{\text{gain}}$ and the measurement noise reduction, with a sharper gain for small reductions which diminishes the further the noise is decreased. Similarly, SNR$_0$ affects the gain, with noisier records benefiting the most. As a test of our analytical solution (Eqn. 16), we also compute $f_{\text{gain}}$ numerically and obtain results matching the analytical method. This validates our numerical approach and establishes confidence in using it for the more complex scenario where $P_0$ has some frequency dependence, where no analytical solution for frequency is possible.





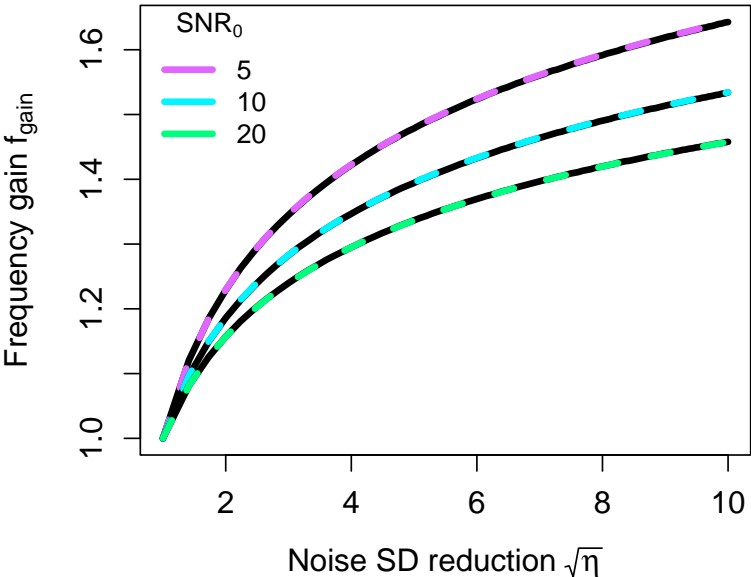

**Figure 4.** Relative gain in frequency $f_{\text{gain}}$ as a function of measurement noise reduction factor ($\sqrt{\eta}$, thus measured in standard deviation), estimated for the white noise case assuming three different initial signal-to-noise ratios (SNR$_0$, i.e. the ratio between the initial signal $P_0$ and the measurement noise level $P_n$). The analytical results (dashed lines) match the corresponding numerical results (solid black lines), corroborating the numerical method.

## 4.2 Application to Beyond EPICA - Oldest Ice Core

Deeper ice cores spanning longer timescales store isotopic data that are more complex than white-noise. In these cases, too many parameters are involved to generalise the measurement noise and highest resolvable frequency relationship, so we instead find estimates through direct application to a specific ice core. Taking the example of the BE-OIC, we calculate absolute $f_{\text{max\_diff}}$ values throughout the record using our diffusion length profile, transformed into time units using the age-depth model, and $A_{\text{diff\_min}} = 0.25$ (dotted lines in Fig. 5a). Similarly, we also calculate $f_{\text{max\_decon}}$ values at the same ages for various mea-

surement noise levels, giving a measurement noise and $f_{\text{max\_decon}}$ relationship predicted for different ages of ice in the BE-OIC (solid lines in Fig. 5a). For a closer inspection, we rescale the x-axis by dividing all $f_{\text{max\_decon}}$ values by the $f_{\text{max\_diff}}$ at the corresponding ages. This presents the relative improvement offered by Wiener deconvolution for different measurement noise levels (Fig 5b). We see the resolvable timescales for a standard 0.1‰ of $\delta^{18}$O measurement noise vary from 5 years (5 kyr old ice) to longer than 10,000 years (1.5 Myr old ice). Throughout the ice core, Wiener deconvolution increases $f_{\text{max}}$ by a factor

of 1.0 to 1.6. Further improvement is seen if the measurement noise is reduced to 0.01‰, with frequencies resolvable up to twice as high as the non-deconvolved record.





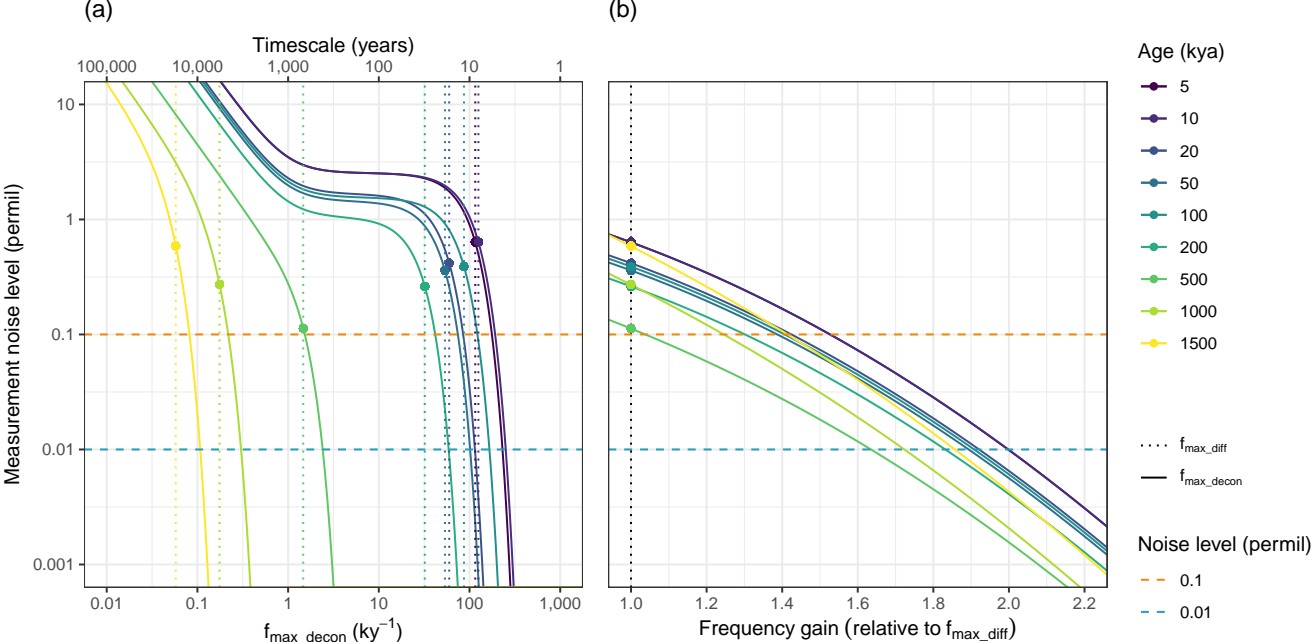

**Figure 5. (a)** Maximum permitted measurement noise to recover the highest resolvable frequency $f_{\mathrm{max\_decon}}$ for selected ages of the BE-OIC after deconvolution, using $A_{\min} = 0.25$. Also shown are the highest resolvable frequencies with no deconvolution $f_{\mathrm{max\_diff}}$ (vertical dotted lines). For improved visibility, the points at which the $f_{\mathrm{max\_diff}}$ lines intersect the corresponding $f_{\mathrm{max\_decon}}$ lines are marked by dots. The horizontal dashed lines represent a typical measurement noise of 0.1‰ (orange) and a 10 times reduction (blue). **(b)** Same as (a) but showing the frequency gain relative to the corresponding $f_{\mathrm{max\_diff}}$ (no deconvolution).

### 4.3 Surrogate ice core record examples

While the highest resolvable frequency is a useful quantification parameter, visualising the direct improvement to a decon-
volved depth profile can provide a more intuitive understanding of the significance. Using our fit to the average Dome C
spectrum (Fig. 2) as an initial climate signal, $P_0$, we simulate two isotopic records at a 5 cm resolution representing the two
different climate states which make up our model (Eqn. 1). The firn/upper ice scenario, where the variability is dominated by
white-noise, has $P_0 = \epsilon$, while the deep ice scenario with temporally correlated climate variability has $P_0 = \alpha f^{-\beta}$. Both simu-
lated records are diffused, using diffusion lengths of $\sigma = 0.08$ m and $\sigma = 0.21$ m for the shallow and deep records respectively,
taken at ages 1 kya and 1.2 Mya from the modelled profile (Fig. B1). Duplicating each record, we add normally distributed
random variations with a standard deviation of $0.1‰$ and $0.01‰$, representing measurement noise. We compare the original
climate signal input to these two surrogate isotope records (Fig. 6a, Fig. 7a) and their Wiener deconvolutions (Fig. 6b, Fig. 7b).
While the surrogate records look similar despite differing noise levels, the improvement in the deconvolution is more apparent,
with the less noisy record better representing the true, original signal. Some features occurring on timescales only recoverable
with the lower measurement noise are restored much more accurately for the less noisy records (e.g. 56 m - 57.5 m in Fig 6b,





2489 m - 2491 m in Fig. 7b). In these examples, the residuals between the original and deconvolved signals (Fig. 6c, Fig. 7c) show a mean reduction of 33% and 49% respectively for the less noisy record.

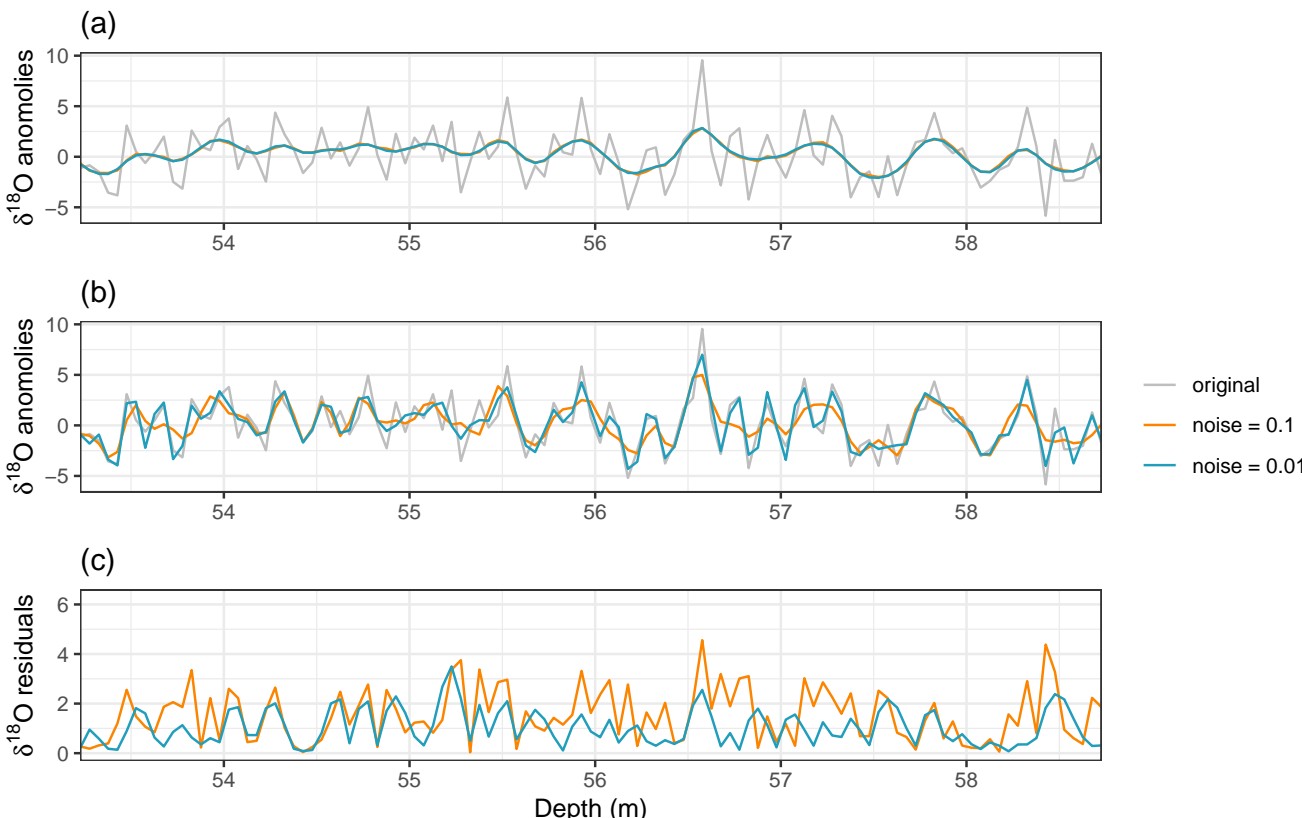

**Figure 6.** Example of the effect of the measurement noise level on the deconvolution in the depth domain for shallow ice. **(a)** The true, undiffused record (grey) and two diffused records ($\sigma = 0.08$ m) with measurement noise standard deviations of 0.1‰ (orange) and 0.01‰ (blue). **(b)** Deconvolutions of the two diffused records in (a). **(c)** Residuals between the original and deconvolved records.





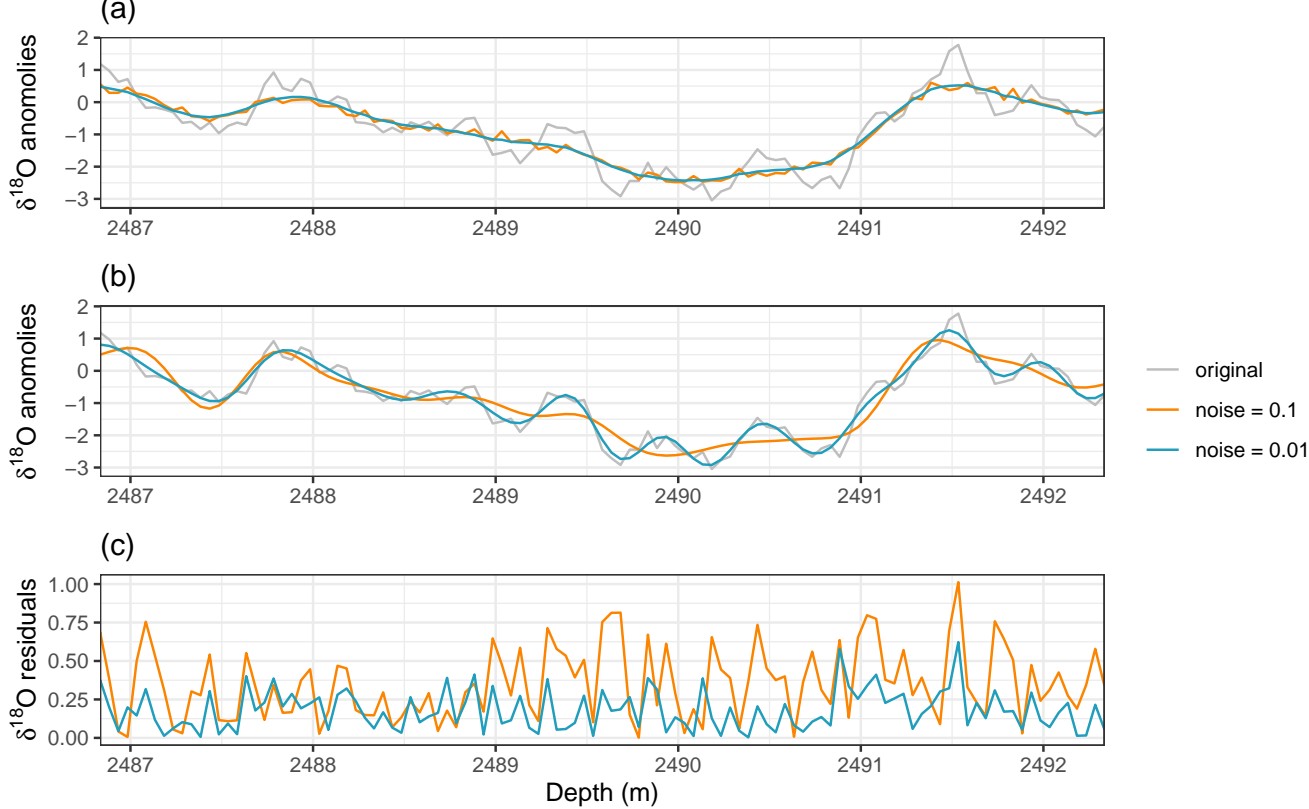

**Figure 7.** Example of the effect of the measurement noise level on the deconvolution in the depth domain for deep ice. **(a)** The true, undiffused record (grey) and two diffused records ($\sigma = 0.15$ m) with measurement noise standard deviations of 0.1‰ (orange) and 0.01‰ (blue). **(b)** Deconvolutions of the two diffused records in (a). **(c)** Residuals between the original and deconvolved records.

## 5   Discussion

This study explores the role of measurement precision in restoring high frequency variability of diffused water isotope signals from ice cores. We achieve this by studying how the measurement noise level affects the maximum recoverable frequency after deconvolution. For this, we use the criterion where Wiener deconvolution is able to restore to at least 25% of a given frequency's amplitude before diffusion. For records spanning shorter time periods, where the undiffused isotopic record can be approximated as white-noise, we derive a closed form expression for the dependency on the measurement noise. We find a non-

linear relationship between measurement noise and the highest resolvable frequency $f_{\mathrm{max\_decon}}$. This relationship also depends on the ratio between the power spectra of the initial signal and the measurement noise. Over the longer timescales covered in deep ice, where the white noise assumption for the initial signal does not hold, we instead preform numerical calculations through direct application to a specific ice core. We use the BE-OIC as an example, estimating absolute frequency limits for different measurement noise levels. Additionally, we show the resolution improvement offered by Wiener deconvolution





relative to the diffused, non-deconvolved case, finding similar gains from reducing measurement noise as the generalised white-noise case throughout the ice core.

### 5.1 Dependency on the relative amplitude

Our paper describes a method of quantifying the resolution limit $f_{\max}$ imposed by measurement noise. For this aim, a definition is needed for what amplitude loss is still acceptable to the researcher. Here, we choose a relative amplitude $A_{\min} = 0.25$, i.e.

25% of the initial amplitude remaining, since we are interested in the restoration of the climate signal. One may require restoration closer to the initial amplitude, for which a higher minimum relative amplitude can be chosen, say $A_{\min} = 0.9$. Conversely, one could select a lower relative amplitude for a more optimistic $f_{\max}$, although the extra frequencies will have minimal restoration and contain mostly measurement noise. For different relative amplitudes, the response of the corresponding $f_{\max}$ to changes in measurement noise varies (see Appendix C). Predictably, choosing $A_{\min} = 0.05$ slightly increases $f_{\max}$

values, but also reduces the relative frequency gain to approximately $1.4\times$ for $0.01‰$ noise, and even results in a smaller $f_{\mathrm{max\_decon}}$ than $f_{\mathrm{max\_diff}}$ for a noise $0.1‰$ (Fig. C1). In contrast, setting $A_{\min} = 0.9$ demonstrates a much larger frequency gain after Wiener deconvolution, between $1.5\times$ and $4\times$ $f_{\mathrm{max\_diff}}$ for $0.1‰$ noise and from $4.5\times$ to $6\times$ for noise of $0.01‰$ (Fig. C2). This shows that the benefit of reducing measurement noise is greater when striving for more accurate restorations of a signal's original amplitude.

### 5.2 Alternative deconvolution methods

Our results depend on the choice of deconvolution method, for which we have used the Wiener filter. One might ask if there are other deconvolution methods that can recover higher frequencies to a greater extent. Any linear response function will damp or amplify signal and noise equally, and thus not change the SNR. In principle, one could amplify higher frequencies than the Wiener filter allows, and thus get a deconvolved record with more high frequency variability that will be much noisier.

Alternatively, a more conservative reconstruction can be made to keep the average SNR down, but in turn more high frequencies will be lost. The Wiener deconvolution method was selected as it is optimal in terms of minimising the root mean square error, and also due to its prominence in previous water isotope research (Johnsen, 1977; Johnsen et al., 2000; Gkinis et al., 2011) and other paleoclimatic proxy reconstructions (Liu et al., 2021).

### 5.3 Implications for the BE-OIC project

Our study shows that the relative frequency gain through Wiener deconvolution for a measurement noise of $0.1‰$ is between 1.0 and 1.6, and increases to 1.6 - 2.0 for $0.01‰$ of noise. The surrogate data examples illustrate how this difference offered by the noise reduction could be crucial for more accurate representation of deep ice records especially. If features of interest are expected to have high variability over timescales which benefit most from reduced measurement noise, then it is necessary to measure with enough precision to fully capture the important features. For example, one aim of the Beyond EPICA project is

to recover glacial-interglacial patterns before the Mid-Pleistocene Transition, which have a periodicity of 41 kyr. Our estimate





in Fig. 5 shows a measurement noise of 0.1 ‰ has an $f_{\max}$ of < 0.1 kyr$^{-1}$ for 1.5 Myr old ice. To recover features like the asymmetry of glacial-interglacial cycles, or overshoots after deglaciation, information on frequencies > 0.1 kyr$^{-1}$ could be necessary, and thus reducing the noise may prove very beneficial.

It is also worth comparing our maximum resolvable frequency of 0.1 kyr$^{-1}$ for the oldest ice with previous predictions. Fischer et al. (2013) considers ice with age densities > 20 kyr m$^{-1}$ to no longer contain any measurable paleoclimate signal, justified as the maximum density after which greenhouse gases have potentially been homogenised by diffusion. Our limit defined and discussed in this paper concerns the diffusion and recovery of stable water isotopes, for which we predict a more optimistic threshold of up to 10 kyr cycles in the deepest ice. We therefore argue that while there are clear limitations on the

gain achieved by reducing measurement noise, given the value and difficulty of retrieving such an old proxy record, acquiring high measurement precision is still of great importance to achieve the most paleoclimatic information possible from the oldest ice core, especially for the previously unextracted and precious >1 Myr old ice.

### 5.4    Future directions - How to decrease measurement noise

Reducing the $\delta^{18}$O measurement noise to the 0.01‰ suggested in this paper, or even further, is already achievable through multiple advances in the area. Firstly, the L2140-i Cavity Ring-Down Spectroscopy water isotope analyser from PICARRO now offers a $\delta^{18}$O measurement noise level of 0.025‰. Additionally, PICARRO claim using the high precision mode, with long integration time measurements of 54 minutes per sample, the measurement noise can be reduced to 0.01‰. Indeed, Steen-Larsen and Zannoni (2024) demonstrates even better measurement precision is possible, with Allan deviations as low

as 0.005‰ for $\delta^{18}$O after measurement times on the order of 2 - 3 hours per sample. While this is undoubtedly a long time to measure each sample, it may still be considered appropriate for the deepest ice in the BE-OIC. Since the highly sought-after and previously unrecovered time period of 800 - 1500 kya spans only the bottom ~75 m (Chung et al., 2023), there are very few samples which would require such precision, and therefore the time commitment is less significant.

Another factor to consider is how the sampling resolution of discrete records affects the highest resolvable frequency. In strongly diffused deep ice, the additional high frequencies acquired through finer sampling are irrelevant for climate reconstructions as the measurement noise will dominate the signal. However, the data can be averaged in time to the desired frequency. If the measurement noise is uncorrelated, this reduces the measurement noise by a factor of $\sqrt{N}$ where N is the number of data points per bin. This in turn has an effect on the highest frequency for which the climate signal is recoverable. However,

given the greatly increased time commitment necessary for cutting and measuring at a reduced discrete sampling size, an improvement of $\frac{1}{\sqrt{N}}$ is not beneficial enough to advocate over alternative, more direct methods of reducing measurement noise. Alternatively, taking duplicate cores also allows any white measurement noise to be reduced by a factor of $\sqrt{N}$, where N is the number of cores. This in turn improves the SNR and enables higher frequencies to be recovered using Wiener deconvolution, but can be a difficult task for larger, deep ice core projects. Fundamentally, given the extra effort required, the ideal precision



comes down to a trade-off between the improvement of the deconvolved record and the additional investment needed to obtain it.

## 6   Conclusions

In conclusion, we define the highest resolvable frequency $f_{\max}$ of water isotope ice core records, both before and after deconvolution, and analyse the limitation introduced by measurement noise for the deconvolved records. We find that enhancing
measurement precision leads to a non-linear increase in $f_{\max}$, with a similar gain across various isotopic signals expected in Antarctic ice cores. For records spanning up to centennial timescales, a general increase in $f_{\max}$ of approximately $1.5\times$ can be achieved by decreasing the measurement noise by an order of magnitude. While difficult to generalise for ice cores covering millennial or longer timescales, due to more complex spectral properties, we apply our approach to the Beyond EPICA Oldest Ice Core (BE-OIC), whose deep ice is yet to be measured. Under our definitions, we see an $f_{\max}$ before diffusion for 1.5
Mya ice of approximately $0.05$ kyr$^{-1}$, with measurement noise below $0.02$‰ required to recover sub-10,000 year variability after Wiener deconvolution. Throughout the ice core, we find Wiener deconvolution with a standard $\delta^{18}$O measurement noise of $0.1$‰ offers an $f_{\max}$ increase of about $1.4\times$ the uncorrected record, which increases up to $2\times$ for a measurement noise of $0.01$‰. Ultimately, judging the value of improved measurement precision comes down to a case-by-case basis, weighing up the frequency gain against the extra effort required. For example, in the case of the BE-OIC, where layer thinning could
result in orbital frequencies approaching $f_{\max}$, it is arguably worth the extra precision, as the additional effort would only need applying to the few samples spanning >800 kya ice. With the advancement of measuring capabilities, the ice core community is already progressing towards improved precision, which will benefit the BE-OIC and future deep ice core projects.

*Code and data availability.* The water isotope data used in this study was previously published and can be found at: https://doi.org/10.1594/ PANGAEA.939445 (Gkinis et al., 2021). The age-depth model for BE-OIC was previously published and was obtained through personal
communication (14.03.2024). Code used in this study is available upon request

## Appendix A:  Composite Dome C Spectra and Diffusion Lengths

The EDC $\delta^{18}$O data is shown in Fig A1 and the corresponding power spectra in Fig. A2. The mean spectrum was estimated by interpolating each power spectrum to the highest frequency spacing (the green spectrum) and then averaging for each frequency.



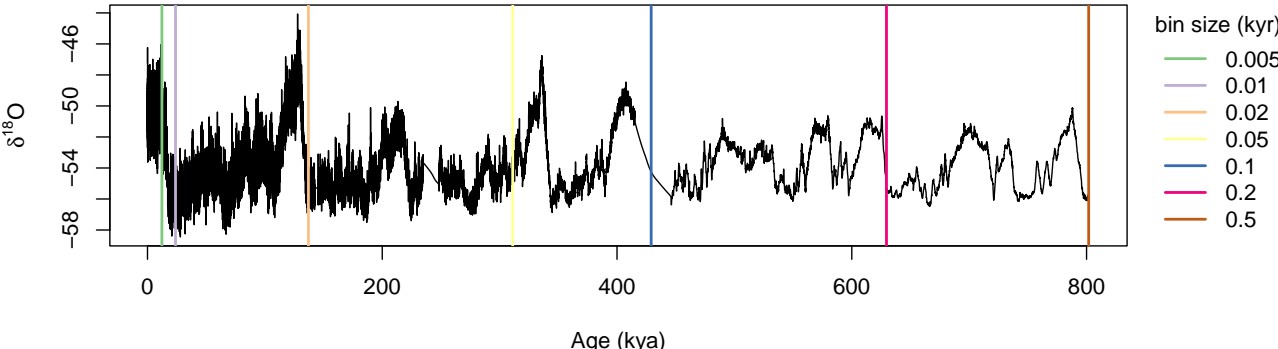

**Figure A1.** Water isotope data from the EDC core used to estimate the local climate spectrum. The data was binned at different timescales, from a depth of 0 m until the depth at which the sample spacing was larger than the given bin size. The vertical lines mark the deepest data point of the corresponding bin size.

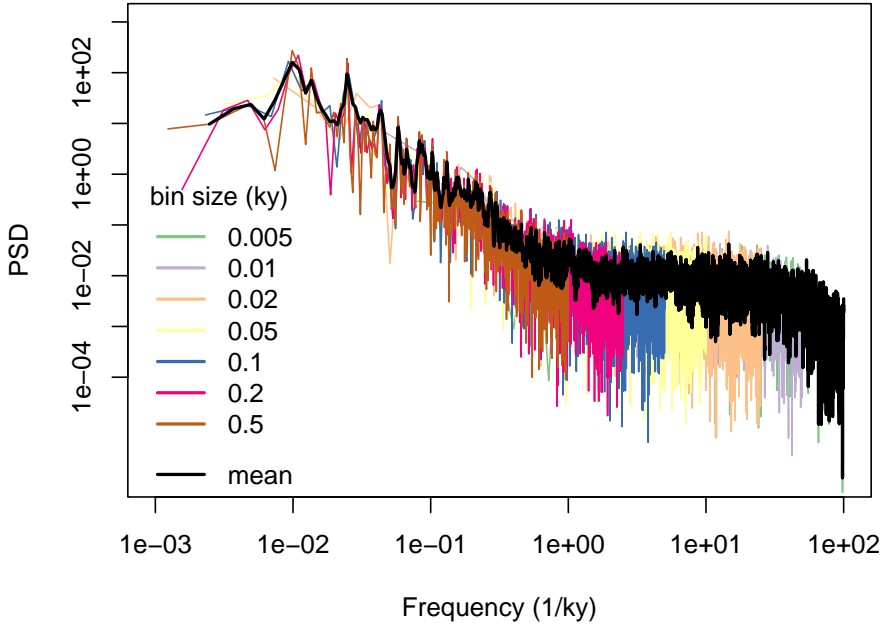

**Figure A2.** Power spectra of the timeseries shown in Fig. A1. The mean of all the power spectra is shown in black.





## Appendix B: Diffusion length calculation

Here we provide a first order calculation of the diffusion length for the BE-OIC. The latter is the result of the diffusion in the firn pore space and the solid diffusion within the ice grain boundaries (Gkinis et al., 2014; Holme et al., 2018). The first process ceases at the pore close-off while the latter persists down to the bottom of the core. The total signal as a function of depth $z$ is described by (Gkinis et al., 2014)

$$\sigma^2(z) = [S(z)\sigma_{\text{firn}}(z)]^2 + \sigma_{\text{ice}}^2(z) \tag{B1}$$

where $S(z)$ is the ice flow thinning.

### B1 The firn diffusion calculation

We use a depth domain spanning from the surface (z = 0) to a depth of 2579 m based on the results from Lilien et al. (2021). The latter refers to the depth at which the flowing ice slab meets the stagnant basal unit at the LDC site. For the calculation of the $\sigma_{\text{firn}}$ quantity we use the analytical solutions described in (Gkinis et al., 2021) using:

- Surface temperature $T_0 = 213.15\ \text{K}$

- Accumulation $A = 0.016\ \text{ma}^{-1}$ ice equivalent

- Surface Density $\rho_0 = 330\ \text{kgm}^{-3}$

- Close-off density $\rho_{\text{co}} = 804\ \text{kgm}^{-3}$

- Atmospheric pressure $P = 0.6\ \text{Atm}$

### B2 The thinning function

For the calculation of the thinning function we use the Lliboutry model (Lliboutry, 1979) for the annual layer thickness described by:

$$\lambda(z) = A\left[1 - \frac{p+2}{p+1}\left(\frac{z}{H}\right) + \frac{1}{p+1}\left(\frac{z}{H}\right)^{p+2}\right] \tag{B2}$$

where the shape factor $p = 5.5$ and the ice thickness $H = 2579$ (Lilien et al., 2021). A steady state time scale is then calculated from Eq. B2 as:

$$\bar{t} = \int_0^z (\lambda(z'))^{-1}\ \mathrm{d}z' \tag{B3}$$

We follow the approach shown in Parrenin et al. (2017); Lilien et al. (2021) introducing a change of variable in a pseudo-steady scheme as:

$$\bar{t} = \int_0^t \frac{A_{\text{edc}}(t')}{\bar{A}_{\text{edc}}(t')}\ \mathrm{d}t', \tag{B4}$$



where $A_{\text{edc}}$ is the accumulation at the Dome C site as a function of time (Bazin et al., 2013). We solve Eq. B4 for $t$ using the Brent algorithm (Brent, 1973) implemented in the Scipy toolbox (Jones et al., 2001) and obtain a timescale adjusted for a time dependent accumulation. For ages older than the bottom of EDC at 803 ka the normalized accumulation is assumed to be equal to 1.

**B3    The ice diffusion calculation**

For the calculation of $\sigma_{\text{ice}}^2(z)$ we integrate the ice diffusivity in the time domain taking into account the effect of ice flow thinning (Gkinis et al., 2014; Holme et al., 2018):

$$\sigma_{\text{ice}}^2(t') = S^2(t') \int\limits_0^{t'} 2D_{\text{ice}}(t)S(t)^{-2}\mathrm{d}t \tag{B5}$$

We use the Ramsaeir parametrization for the diffusivity coefficient in solid ice (Ramseier, 1967):

$$D_{\text{ice}} = 9.2 \cdot 10^{-4} \cdot \exp\left(-\frac{7186}{T}\right)\ \mathrm{m^2 s^{-1}} \tag{B6}$$

**B4    Temperature profile calculation**

We calculate the temperature profile by solving the parabolic differential equation accounting for advection using an inverted depth domain ie distance from top of the stagnant ice slab:

$$\frac{\partial T}{\partial t} = D_{\text{T}}\frac{\partial^2 T}{\partial h^2} + w\frac{\partial T}{\partial h} \tag{B7}$$

where $D_{\text{T}}$ is the thermal diffusivity coefficient ($1.13 \cdot 10^{-6}$) and the advection coefficient $w = -\lambda$. We use a automated finite difference scheme using the method of lines utilizing the Scikit-FDiff package (Cellier and Ruyer-Quil, 2019). The model run spans $10^6$ a with a time step of 100 a and a spatial resolution of 10 m.

We use a Dirichlet boundary condition for the surface with $T_0 = 213.15\ \mathrm{K}$ and a Neumann boundary condition for the boundary between the stratified ice and the stagnant ice. The initial profile is linear with $T_0$ at the top and a bottom temperature
at 272 K. Choosing the value of the Geothermal Heat Flux is a question that deserves a dedicated study. For this tentative calculation we have chosen $G = 2.7 \cdot 10^{-2}\ \mathrm{Wm^{-2}}$. This value lies in the low range of the spectrum indicated by Van Liefferinge et al. (2018), assuming though that we are omitting the stagnant ice slab with a thickness of $\approx 200\ \mathrm{m}$ this value is a reasonable choice for a first order calculation of the temperature profile and the ice diffusion length thereafter. Further consideration of the temperature profile when the borehole is logged will obviously allow for a better estimation of the diffusion at the bottom
of the BE-OIC.





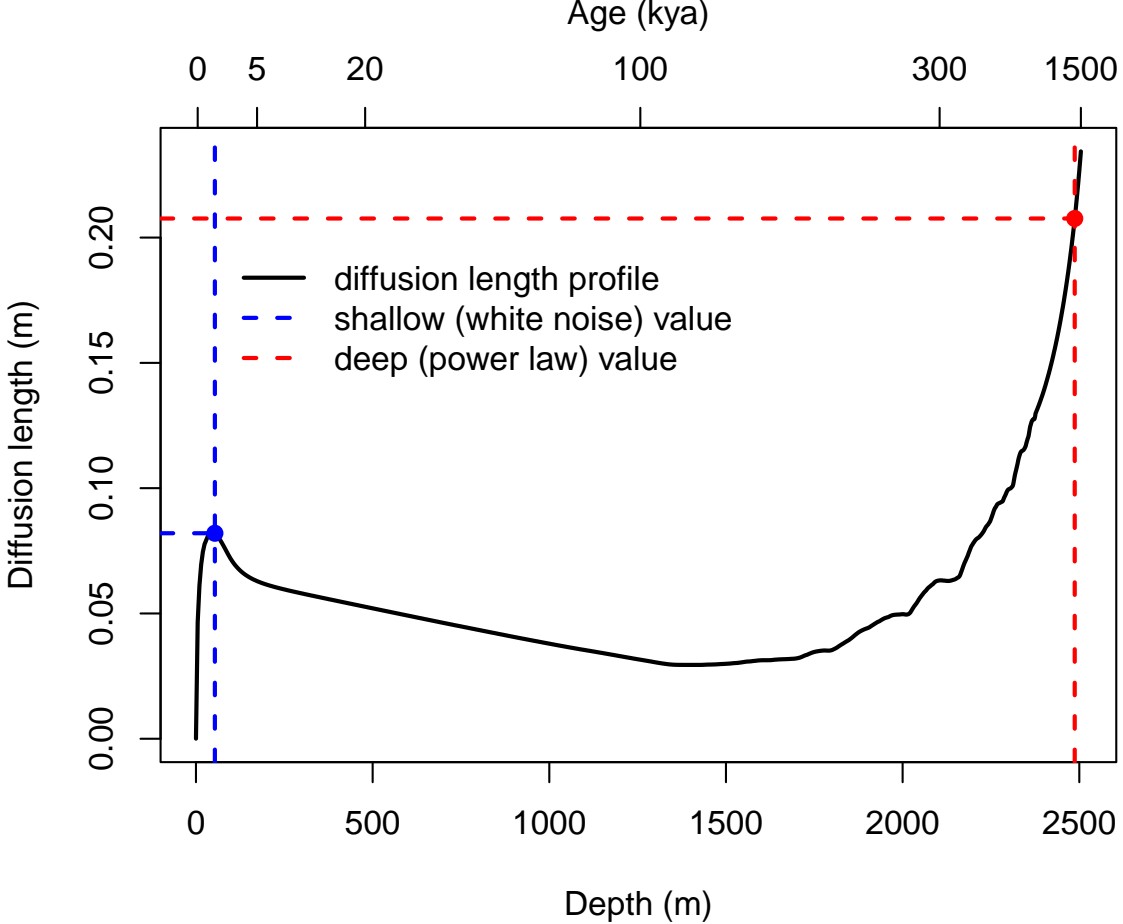

**Figure B1.** Modelled diffusion length profile for the BE-OIC (black). The diffusion lengths used in the simulated isotopic records in Section 4.3 were selected from the profile (white-noise case is red, power-law case is blue).





## Appendix C: Sensitivity to $A_{\min}$

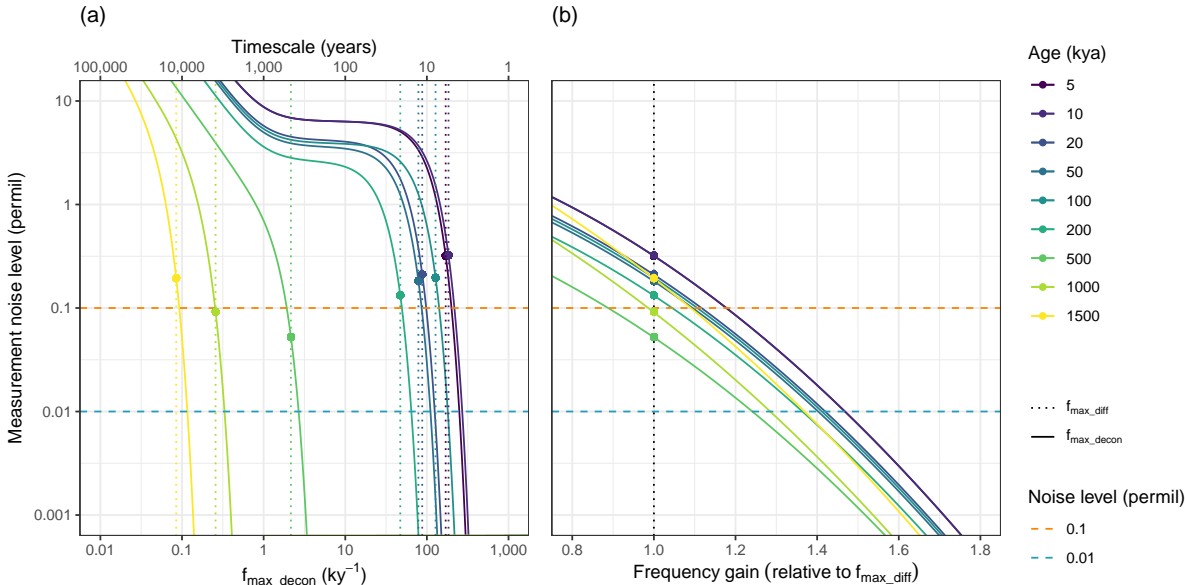

**Figure C1.** Same as Fig. 5 but with $A_{\min} = 0.05$.

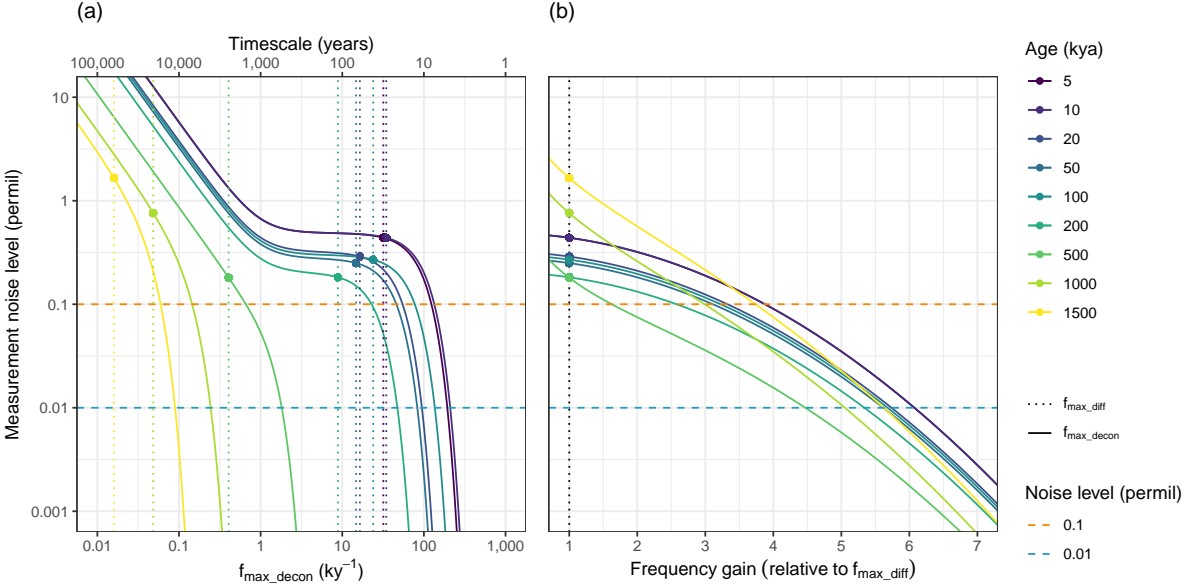

**Figure C2.** Same as Fig. 5 but with $A_{\min} = 0.9$.



*Author contributions.* TL and FS designed the study. TM and TL contributed to the analytical derivations and illustration of results. VG provided the modelled diffusion length data. FS performed the analysis and wrote the manuscript. All authors contributed to the discussion of the results and to the preparation of the final manuscript.

*Competing interests.* The contact author has declared that none of the authors has any competing interests.

*Acknowledgements.* This publication was generated in the frame of the DEEPICE project and received funding from the European Union's Horizon 2020 research and innovation programme under the Marie Skłodowska-Curie grant agreement no. 955750 and the ERC SPACE no. 716092. It contributes to Beyond EPICA, which received funding from the European Union's Horizon 2020 research and innovation programme under grant agreement no. 815384 (Oldest Ice Core). Thomas Münch was supported by the Informationsinfrastrukturen Grant

of the Helmholtz Association as part of the DataHub of the Research Field Earth and Environment. Vasileios Gkinis was supported by the Villum Foundation ("The whisper of ancient air bubbles in polar ice", grant no. 00028061, and "Unraveling paleo-climate knots with lasers", grant no. 00022995). This is Beyond EPICA publication number XX.



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
