# Peer review of "The impact of measurement precision on the resolvable resolution of ice core water isotope reconstructions"

_EGUsphere, 2024_

## Author Comment (AC1)

**Response to Reviewer 1 (author's comments in blue)**

This manuscript presents a well-structured and insightful study on the role of measurement precision in enhancing the resolvable resolution of ice core water isotope reconstructions, particularly focusing on the Beyond EPICA Oldest Ice Core (BE-OIC). The research is timely and addresses a critical gap in paleoclimate reconstructions by quantifying how measurement noise limits the recovery of high-frequency climate signals through deconvolution techniques. The analytical and numerical approaches are robust, and the results are clearly presented. The manuscript is suitable for publication after minor revisions.

We thank the reviewer for their recognition of the importance of the work and their overall positive assessment of the manuscript.

The description of the Wiener deconvolution process and its application to ice core data is technically sound but could be more accessible. Figure 1 shows the conception of how reducing the measurement noise to recovery of higher frequencies. I think the legend of figure 1 should be explained in more details to aid readers unfamiliar with signal processing.

We have modified the deconvolution explanation in lines 32 - 45 to with clearer and more precise language. We have also added the following description of Figure 1 at the end of the paragraph in order to aid the reader with the concept:

Fig. 1 illustrates a simplified version of this concept through idealised power spectra. The variability of the diffused signal (black) decreases with increasing frequency, until it falls below the variability of the measurement noise (dashed lines). The corresponding frequencies at which SNR = 1 is marked by the vertical lines. It is clear to see that decreasing the measurement noise by some amount $\Delta P_n$ (orange to blue) increases the frequency at which SNR = 1 by $\Delta f$.

Line 69-73, the authors indicate that the post-depositional processes have a lesser effect on longer timescales. However, the diffusion process is one of the post-depositional processes. Here is a little confused. What kind of the post-depositional processes do you indicate?

Here we were referring to wind redistribution, sublimation, melting, etc. which are more accurately described as depositional processes, not post-depositional. This has been corrected.

The diffusion length profile (Fig. B1) is critical to the analysis but is only briefly mentioned. More details on its derivation, including sensitivity to parameters like geothermal heat flux, would strengthen the manuscript.

In the revised version, we will expand the description of the diffusion length profile and include a discussion of how the results are affected by its uncertainty. However, the main focus of the manuscript is the influence of measurement precision on the achievable

resolution in ice core water isotope reconstructions. These insights are broadly applicable, rather than tied to a specific ice core or diffusion length profile. The BE-OIC case serves as one illustrative example. Therefore, a comprehensive treatment of the derivation and all associated uncertainties of the diffusion length profile, including sensitivity to parameters such as geothermal heat flux, would exceed the scope of this study.

The authors should discuss the limitations of the deconvolution method. For example, briefly acknowledge any assumptions in the Wiener deconvolution method that might affect real-world applicability (e.g., linearity of diffusion, stationarity of noise) and discuss potential biases or uncertainties in the diffusion length profile (Appendix B) and how they might influence the results.

We've added a subsection to the discussion which details the following assumptions and application limitations:

One assumption made when applying the Wiener deconvolution method across some time series segment is that the diffusion length remains constant throughout the segment. In reality, since the diffusion length varies with depth, the mean diffusion length across the segment is used, resulting in slightly biased reconstructions. In most cases, this error can be minimised by applying the deconvolution across smaller segments.

Throughout the paper, measurement noise was approximated as white-noise, which enabled more generalisation and simplified some calculations. This approximation is appropriate for discretely sampled data, such as the Dome C water isotope record, due to the independence of consecutive measurements. However, when working with isotopic data from continuous flow analysis measurements, the complications from additional instrumental memory effects should be considered.

Given the limited data, the diffusion length model of the oldest ice core used in this study makes some necessary assumptions and thus the profile has significant uncertainty. While these uncertainties impact the absolute values of $f_{max}$ calculated at each depth/age, the relative gain in reducing measurement noise remains largely unaffected. As such, the key point which quantifies the importance of reducing measurement noise is still valid.

Finally, in this study the PSD before diffusion ($P_0$) for the oldest ice core was estimated from the Dome C water isotope record. The convenience of an equivalent ice core from which $P_0$ can be estimated before measuring the new core will not always be available, especially for deep ice cores, and as such the $P_0$ estimation could be more challenging in future applications of this method.

---

## Author Comment (AC2)

**Response to Reviewer 2 (author's comments in blue)**

Shaw et al report an analytical method to quantify the upper limit of recoverable frequency from diffused ice core isotopic records after deconvolution, with a particular focus on the effect of measurement noise. The authors suggest that the upper part can be analytically solved where white-noise dominates. Then the authors apply the numerical solutions to the Beyond EPICA Oldest Ice Core, for which climate reconstruction is still ongoing. Overall, the study provides valuable insights into the recovery of water isotope signals in deep ice cores, especially the usefulness of the Beyond EPICA Oldest Ice Core in reconstructing climate before the Mid-Pleistocene Transition. I have a few minor comments on the manuscript.

We thank the reviewer for the positive feedback and the emphasis on the current relevance of our work.

Although the authors provide two surrogate ice core record examples, it would be helpful to generalize how reducing the measurement noise improves the deconvolved depth profile. For instance, only two different diffusion lengths are discussed in the study. How would different diffusion lengths influence the results?

For the white-noise initial PSD ($P_0$) case, the frequency gain could be generalised as it only depends on the measurement noise reduction and the initial signal-to-noise ratio, and is independent of the diffusion length, as shown in Figure 4. For more complex $P_0$s, generalising is difficult, due to the dependence on two additional factors (diffusion length and $P_0(f)$). The BE-OIC plots show the result for 9 cases, with each different coloured line representing a different Age with varying diffusion lengths and $P_0$s. Figures 6 and 7 offer a more qualitative picture of the potential improvement in the recovery of the isotopic data, with a specific example for both the shallow ice and deep ice cases.

To give a more general picture of the effect of measurement noise on the recovered depth series, we compute the residuals between the true and deconvolved isotopic records for both the high noise (0.1 permil) and low noise (0.01 permil) cases (as in Figures 6c and 7c). We can then find the ratio of the residuals between the two noise cases. Here, for example, a ratio of 2 means the 0.1 permil has double the mean residuals than the 0.01 permil case. Repeating this across different diffusion lengths, with 100 realisations per diffusion length, gives the following figures:

[Figure]

Here, the vertical dashed lines represent the diffusion lengths used in Figure 6 and 7, and the diffusion lengths range from half to double these values. The lower diffusion length values for the upper ice case show a much improved depth profile for the 0.01 permil record. This is expected, as a very small diffusion length (<5 cm) results in a less sharp decrease in power for the higher frequencies, and thus frequencies can be gained at a faster rate when reducing the measurement noise. There is minimal change in the ratio for the deep ice case.

Please check the consistency of the mathematical expressions throughout the manuscript. For instance, $P_n$ and $P_n(f)$ are used interchangeably.

The inconsistent notation was corrected. All power spectra terms are now initially written with frequency dependence. When assuming a constant white-noise power spectrum for a given term, the notation without frequency dependence is used for simplicity, and is now explicitly stated when used for the first time.

L86-87: it would be beneficial to provide evidence or justification for these choices?

The lowest frequencies are excluded from the fit due to the known bias in the spectral estimation method (Thomson, D. J. (1982). Spectrum estimation and harmonic analysis. Proceedings of the IEEE, 70(9), 1055–1096. https://doi.org/10.1109/PROC.1982.12433). We have added this justification and reference to the text.

For the upper frequency cut-off, the firn diffusion is evident from the power spectrum. It can be justified mathematically by taking the diffusion length and layer thickness in the firn and showing the loss of amplitude at frequencies above 10 kyr $^{-1}$.

At a depth of 30 m we can expect a diffusion length of $\sigma \sim = 0.08$ m

For the Dome C site, the layer thickness at 30 m depth is 42 m kyr $^{-1}$

Thus the frequency in m$^{-1}$ corresponding to 10 kyr $^{-1}$ at this depth is:

$$f_{depth} = 10/42 = 0.24m^{-1}$$

Therefore, the remaining power of the 10 kyr $^{-1}$ frequency is equal to:

$$P_{remaining} = e^{-(2\pi\sigma f_{depth})^2} = e^{-(2\pi * 0.08 * 0.24)^2} = 0.986$$

Or a power loss of 1.4 %. Any frequencies at a faster timescale than this will have lost even more power to firn diffusion, and are therefore not appropriate for our analogue of the isotopic record before diffusion, $P_0(f)$.

We have added the above explanation to Appendix A.

Figure 3b: what is the purple curve?

The purple curve is the Wiener filter, as stated in the figure caption.

The diffusion length stated at L184 appears to differ from that in the caption for Figure 7.

Well spotted, this is a mistake, it has been corrected.

Figures 6 and 7: it is unclear how "the true, undiffused" records were obtained.

The "true, undiffused" records are stochastic depth series. They are generated by first creating white noise, scaling its Fourier transform to match the target power spectrum, and then applying the inverse Fourier transform to obtain the depth-domain signal. In the shallow ice case (Figure 6), the target power spectrum is simply white-noise with variance equal to the white-noise regime in Figure 2 ($\epsilon$ in Equation 1). For the deep ice case (Figure 7), the power spectrum is rescaled to a power-law, defined by the low frequencies of the Figure 2 fit (given by $\alpha f^{-\beta}$ in Equation 1). The shallow and deep ice power spectra are smoothed by diffusion lengths of 8 cm and 21 cm respectively, following Equation 5. Finally, after taking the inverse Fourier transforms, white-noise representing measurement noise is added to the resulting depth series. This is done with measurement noises with standard deviations of 0.1‰ and 0.01‰, giving the orange and blue depth series respectively (in both Figures 6 and 7).

The text has been edited to better explain this method.